# A New Approach to Backtracking Counterfactual Explanations: A Unified Causal Framework for Efficient Model Interpretability

Pouria Fatemi [1 2]   Ehsan Sharifian [3]   Mohammad Hossein Yassaee [4]

## Abstract

Counterfactual explanations enhance interpretability by identifying alternative inputs that produce different outputs, offering localized insights into model decisions. However, traditional methods often neglect causal relationships, leading to unrealistic examples. While newer approaches integrate causality, they are computationally expensive. To address these challenges, we propose an efficient method called BRACE based on backtracking counterfactuals that incorporates causal reasoning to generate actionable explanations. We first examine the limitations of existing methods and then introduce our novel approach and its features. We also explore the relationship between our method and previous techniques, demonstrating that it generalizes them in specific scenarios. Finally, experiments show that our method provides deeper insights into model outputs.

## 1. Introduction

Machine learning (ML) has become a core technology in areas such as healthcare, finance, and autonomous systems (Bhoi et al., 2024; Xie et al., 2024; Sancaktar et al., 2022). Although ML models are generally very effective, their limited interpretability is still a significant obstacle (Jethani et al., 2021). Understanding *why* a model generates a specific prediction is crucial for trust, fairness, and accountability (Miller, 2019; Zhang & Bareinboim, 2018; Von Kügelgen et al., 2022; Karimi et al., 2023). This need is especially clear in high-stakes domains like medical diagnosis or loan approval, where decisions can lead to serious consequences (Doshi-Velez & Kim, 2017).

Counterfactual explanations are a widely used tool for interpretability. They address two main questions:

1. *"Why did the model produce this outcome?"*
2. *"What changes can lead to a different outcome?"* (Karimi et al., 2022).

These explanations offer localized insights by highlighting minimal modifications to input features that would alter the model's output (Wachter et al., 2017; Karimi et al., 2020a). For instance, in the context of loan applications, a counterfactual explanation could recommend increasing one's income or reducing debt to secure approval.

Despite their benefits, traditional counterfactual methods often overlook causal relationships between features, which can lead to impractical or unrealistic suggestions (Slack et al., 2021). For example, advising someone to lower their income while increasing savings ignores the causal dependency between these factors. This limitation reduces the practical value of such explanations. Causal algorithmic recourse (Karimi et al., 2021) incorporates *Interventional Counterfactuals (ICF)* to produce more realistic outputs, but this approach is typically computationally expensive and difficult to scale.

*Backtracking Counterfactuals (BCF)* (Von Kügelgen et al., 2023) present a new way to define counterfactuals in causal inference. We propose a new framework for generating counterfactual explanations using backtracking counterfactuals. Our method combines causal reasoning with computational efficiency, enabling it to produce actionable explanations at scale. The key contributions of our work are as follows:

- We analyze the limitations of existing counterfactual methods, including their inability to handle causal dependencies and their high computational costs.
- We introduce our novel method, BRACE: Backtracking Recourse and Actionable Counterfactual Explanations, that leverages backtracking counterfactuals to provide actionable and meaningful explanations.
- We show that our new approach unifies existing methods in certain scenarios.
- We demonstrate through experiments that our method provides better insights into model behavior.

[1]Department of Mathematics, Technical University of Munich, Germany [2]Munich Center for Machine Learning, Germany [3]Department of Electrical Engineering, École Polytechnique Fédérale de Lausanne, Switzerland [4]Department of Electrical Engineering, Sharif University of Technology, Tehran, Iran. Correspondence to: Mohammad Hossein Yassaee <yassaee@sharif.edu>.

*Proceedings of the $42^{nd}$ International Conference on Machine Learning*, Vancouver, Canada. PMLR 267, 2025. Copyright 2025 by the author(s).

This paper is organized as follows. Section 2 introduces fundamental concepts in causal inference, counterfactual reasoning, and the problem definition. Section 3 reviews prior work on counterfactual explanations and interpretability. Section 4 details our proposed framework. Section 5 explores the relationship between backtracking and interventional counterfactuals, while Section 6 examines how our method connects to existing approaches. Section 7 discusses metric selection and our optimization method. We present experimental results in Section 8, followed by conclusions and future directions in Sections 9 and 10, respectively.

## 2. Preliminaries and Problem Statement

In this section, we review the notion of Structural Causal Models (SCMs) (Pearl, 2009), discuss interventional versus backtracking counterfactuals, and formally define the problem setting.

### 2.1. Structural Causal Models (SCMs)

A SCM $\mathcal{C} := (\mathbf{S}, P_{\mathbf{U}})$ describes a set $\mathbf{S}$ of causal relationships among variables through structural equations:

$$X_i := f_i(\mathbf{X}_{\text{pa}(i)}, U_i), \quad i = 1, \ldots, n, \tag{1}$$

where $\mathbf{X}_{\text{pa}(i)}$ are the parents variables (direct causes) of $X_i$, and $U_i$ are independent noise terms sampled from a distribution $P_{\mathbf{U}}$. These relationships are represented by a Directed Acyclic Graph (DAG) $G$, which governs the observational distribution $P_{\mathbf{X}}^{\mathcal{C}}$ (Peters et al., 2017).

The acyclic structure of $G$ ensures that each $X_i$ can be expressed as a deterministic function of $\mathbf{U}$. This results in a unique mapping from $\mathbf{U}$ to $\mathbf{X}$, denoted by:

$$\mathbf{X} = \mathbf{F}(\mathbf{U}), \tag{2}$$

commonly referred to as the reduced-form expression. The function $\mathbf{F}(.)$ translates the distribution of latent variables $\mathbf{U}$ into the distribution of observed variables $\mathbf{X}$. We assume causal sufficiency, implying no hidden confounders are present.

Additionally, we adopt a *Bijective Generation Mechanism* (Nasr-Esfahany et al., 2023), which assumes that $f_i(\mathbf{x}_{\text{pa}(i)}, \cdot)$ is invertible for fixed $\mathbf{x}_{\text{pa}(i)}$. This ensures the existence of the inverse mapping $\mathbf{F}^{-1}(.)$, allowing us to recover:

$$\mathbf{U} = \mathbf{F}^{-1}(\mathbf{X}). \tag{3}$$

### 2.2. Interventional and Backtracking Counterfactuals

Let $\mathbf{x}$ be the observed value, and let $\mathbf{x}_{\mathcal{A}}^{\text{CF}} = (x_i^{\text{CF}} : i \in \mathcal{A})$ be an alternative set of values for a subset $\mathcal{A} \subseteq \{1, 2, \ldots, n\}$. A full counterfactual vector $\mathbf{x}^{\text{CF}} =$

$(x_1^{\text{CF}}, x_2^{\text{CF}}, \ldots, x_n^{\text{CF}})$ must agree with $\mathbf{x}_{\mathcal{A}}^{\text{CF}}$ on all indices in $\mathcal{A}$. Intuitively, $\mathbf{x}^{\text{CF}}$ addresses the question: "What would the variables $\mathbf{X}$ have been if $\mathbf{X}_{\mathcal{A}}$ took the values $\mathbf{x}_{\mathcal{A}}^{\text{CF}}$ instead of the observed values $\mathbf{x}_{\mathcal{A}}$?"

We focus on two main ways to form such counterfactuals, both described by random variables $\mathbf{X}^{\text{CF}}$: the *interventional* approach and the *backtracking* approach. Below is a concise explanation of these two methods.

**Interventional Counterfactuals.** In the interventional method, we force the antecedent $\mathbf{x}_{\mathcal{A}}^{\text{CF}}$ by modifying the system's structural functions $\mathbf{S}$ to create a new set $\mathbf{S}^{\text{CF}} = (f_1^{\text{CF}}, f_2^{\text{CF}}, \ldots, f_n^{\text{CF}})$. Specifically, we fix each $f_i^{\text{CF}}$ to be $x_i^{\text{CF}}$ for $i \in \mathcal{A}$, while keeping $f_i^{\text{CF}} = f_i$ for all $i \notin \mathcal{A}$. This process is similar to making a direct change in the causal mechanism of the variables in $\mathcal{A}$, referred to as a hard intervention.

**Backtracking Counterfactuals.** By contrast, backtracking counterfactuals preserve the original structural assignments $\mathbf{S}$ and instead adjust the latent variables $\mathbf{U}$. To enforce $\mathbf{x}_{\mathcal{A}}^{\text{CF}} \neq \mathbf{x}_{\mathcal{A}}$, we introduce a modified set of latent variables $\mathbf{U}^{\text{CF}}$. These are drawn from a backtracking conditional distribution $\mathbb{P}_B(\mathbf{U}^{\text{CF}} \mid \mathbf{U})$ (Von Kügelgen et al., 2023), which controls how closely $\mathbf{U}^{\text{CF}}$ resembles the original $\mathbf{U}$. Once we obtain $\mathbf{U}^{\text{CF}}$, we derive the resulting distribution of $\mathbf{X}^{\text{CF}}$ (given $\mathbf{x}$ and $\mathbf{x}_{\mathcal{A}}^{\text{CF}}$) by marginalizing over all possible values of $\mathbf{U}^{\text{CF}}$.

Both interventional and backtracking perspectives provide valuable insights into counterfactual reasoning but rely on distinct causal reasoning paradigms. Here, we only gave a brief overview of these two approaches. Their precise definitions appear in Appendix A.

### 2.3. Problem Definition

We examine a complex model (e.g., a deep neural network) designed for classification tasks. This model is represented as $h : \mathbb{R}^d \to \{0, 1, \ldots, m\}$, where for a given input $\mathbf{x}$, the model predicts $h(\mathbf{x}) = y$.

The input $\mathbf{x}$ is assumed to follow a SCM $\mathcal{C} = (\mathbf{S}, P_{\mathbf{U}})$, where the structural equations $\mathbf{S}$ are fully known. Formally, if $\mathbf{U}$ denotes the latent (noise) variables of the SCM, the input $\mathbf{X}$ is generated as $\mathbf{X} = \mathbf{F}(\mathbf{U})$, with the function $\mathbf{F}(.)$ explicitly defined. The components of $\mathbf{U}$ are mutually independent and $\mathbf{F}(.)$ is invertible. The goal is to find a counterfactual input $\mathbf{x}^{\text{CF}}$ that satisfies:

1. $\mathbf{x}^{\text{CF}}$ is similar to $\mathbf{x}$,
2. $h(\mathbf{x}^{\text{CF}}) = y^{\text{CF}} \neq y = h(\mathbf{x})$, and
3. the causal structure of the input variables is maintained.

In essence, $y^{\text{CF}}$ represents the desired outcome of the model, and the task is to determine the nearest plausible input that

would produce this outcome. This problem definition sheds light on why the model predicted $y$ instead of $y^{\mathrm{CF}}$ and provides a localized understanding of the model's behavior around $\mathbf{x}$.

## 3. Related Work

Methods for interpretability are generally classified into feature-based and example-based approaches (Molnar, 2020). Feature-based techniques attribute model predictions to input features, offering global or local interpretability. For instance, SHapley Additive exPlanations (SHAP) (Lundberg & Lee, 2017) decompose predictions into additive contributions of features. Extensions like Causal Shapley Values (Heskes et al., 2020) and Asymmetric Shapley Values (Frye et al., 2020) incorporate causal dependencies or relax symmetry assumptions, respectively, to enhance the interpretive granularity and address redundancies. Local surrogate models, such as Local Interpretable Model-Agnostic Explanations (LIME) (Ribeiro et al., 2016), provide localized, model-agnostic explanations by approximating the behavior of black-box models for individual predictions.

Example-based methods focus on understanding models through data points. Prototypes and criticisms (Kim et al., 2016) identify representative and atypical samples, while Contrastive Explanations (Dhurandhar et al., 2018) highlight minimal features that sustain or alter predictions. Counterfactual explanations (Wachter et al., 2017), a prominent example-based approach, aim to find minimal modifications to input features that result in different model outputs. These methods are inherently model-agnostic, localized, and intuitive for decision support systems.

In recent years, causality has played a growing role in interpretability. Causal Algorithmic Recourse (Karimi et al., 2021) generates actionable and realistic counterfactuals by respecting causal structures. This approach ensures the plausibility of counterfactuals by adhering to causal dependencies in the data. Subsequent research has extended this framework to address various challenges. For instance, (Karimi et al., 2020b) weakens the assumption of fully known causal graphs and proposes methods for algorithmic recourse when causal knowledge is incomplete. Similarly, (Dominguez-Olmedo et al., 2022) focuses on generating robust and stable algorithmic recourse by introducing cost functions tailored to ensure resilience against adversarial perturbations. Additionally, advancements like (Janzing et al., 2020) refine feature attributions using causal insights, and (Jung et al., 2022; Wang et al., 2021) explore novel Shapley value formulations incorporating causality to create more meaningful interpretations.

A novel direction involves backtracking counterfactual explanations (Von Kügelgen et al., 2023), which modify latent variables while preserving causal dependencies, thereby ensuring consistency with the structural causal model. This approach has been extended through practical algorithms, such as Deep Backtracking Explanations (Kladny et al., 2024), enabling computation of backtracking counterfactuals in high-dimensional settings.

Our approach belongs to the category of example-based methods, focusing on counterfactual explanations. It is directly comparable to methods such as *Counterfactual Explanations* (Wachter et al., 2017), *Causal Algorithmic Recourse* (Karimi et al., 2021), *Backtracking Counterfactual Explanations* (Von Kügelgen et al., 2023), and *Deep Backtracking Explanations* (Kladny et al., 2024). Below, we briefly review and critique these methods.

**Counterfactual Explanations:** The method in (Wachter et al., 2017) generates counterfactuals through the following optimization:

$$
\begin{aligned}
\arg\min_{\mathbf{x}^{\mathrm{CF}}} \quad & d_X(\mathbf{x}^{\mathrm{CF}}, \mathbf{x}) \\
\text{s.t.} \quad & h(\mathbf{x}^{\mathrm{CF}}) = y^{\mathrm{CF}}
\end{aligned}
\tag{4}
$$

A key drawback of this approach is its failure to account for causal dependencies among input variables, often leading to counterfactuals that are unrealistic or infeasible. For example, in a loan approval scenario, it may suggest decreasing age while increasing education level, violating causal relationships. Although these counterfactuals minimize the distance to the original input, they offer little practical guidance for future improvements and fail to provide actionable insights.

**Causal Algorithmic Recourse:** The method in (Karimi et al., 2021) addresses feasibility by optimizing the following:

$$
\begin{aligned}
\arg\min_{\mathcal{A}} \quad & \mathrm{cost}(\mathcal{A}; \mathbf{x}) \\
\text{s.t.} \quad & h(\mathbf{x}^{\mathrm{CF}}) = y^{\mathrm{CF}} \\
& \mathbf{x}^{\mathrm{CF}} = \mathbf{F}_{\mathcal{A}}\left(\mathbf{F}^{-1}(\mathbf{x})\right)
\end{aligned}
\tag{5}
$$

Here, $\mathrm{cost}(.;\mathbf{x})$ measures the intervention cost, and $\mathbf{F}_{\mathcal{A}}(.)$ represents causal functions after intervening on $\mathcal{A}$. While this method ensures actionable counterfactuals, it has two challenges. First, the optimization is combinatorial, requiring a search over all subsets $\mathcal{A}$, which grows exponentially with $n$ input variables ($2^n$ subsets). Second, the method relies on interventional counterfactuals, which are often criticized for lacking causal intuition (Dorr, 2016). Backtracking counterfactuals are considered a better alternative.

**Backtracking Counterfactual Explanations:** The method in (Von Kügelgen et al., 2023) formulates the problem as:

$$
\arg\max_{\mathbf{x}^{\mathrm{CF}}} \quad \mathbb{P}_B(\mathbf{x}^{\mathrm{CF}} \mid y^{\mathrm{CF}}, \mathbf{x}, y),
\tag{6}
$$

focusing on the backtracking conditional distribution $\mathbb{P}_B(\mathbf{U}^{\text{CF}} \mid \mathbf{U})$, which adjusts latent variables to produce counterfactuals. However, its main drawback is the dependence on $\mathbb{P}_B$. Different choices of this distribution lead to varying counterfactuals, and selecting $\mathbb{P}_B$ is left to the user. Additionally, solving (6) becomes computationally challenging for complex $\mathbb{P}_B$ distributions, as we must integrate over all values of this distribution to compute backtracking counterfactuals (see Appendix A).

**Deep Backtracking Explanations:** The method in (Kladny et al., 2024) refines backtracking counterfactuals using this optimization:

$$\arg \min_{\mathbf{x}^{\text{CF}}} \quad d_U \left( \mathbf{F}^{-1} \left( \mathbf{x}^{\text{CF}} \right), \mathbf{F}^{-1} \left( \mathbf{x} \right) \right)$$
$$\text{s.t.} \quad h(\mathbf{x}^{\text{CF}}) = y^{\text{CF}} \tag{7}$$

This method eliminates dependence on $\mathbb{P}_B(\mathbf{U}^{\text{CF}} \mid \mathbf{U})$ by focusing on the latent space distance $d_U$. However, it ignores proximity between $\mathbf{x}^{\text{CF}}$ and $\mathbf{x}$ in the observed space. As a result, the generated counterfactuals may lack intuitive interpretability and fail to meet the original goal of being close to $\mathbf{x}$.

While these methods offer valuable insights, they have notable limitations. In the next section, we propose a new approach that addresses these issues and provides a more effective solution.

## 4. Our method

In this section, we propose our method called BRACE: Backtracking Recourse and Actionable Counterfactual Explanations. As discussed earlier, one of the main limitations of backtracking counterfactuals is their reliance on the conditional distribution $\mathbb{P}_B(\mathbf{U}^{\text{CF}} \mid \mathbf{U})$. This dependency arises because the choice of $\mathbb{P}_B(\mathbf{U}^{\text{CF}} \mid \mathbf{U})$ significantly influences the resulting counterfactuals, and its specification is left entirely to the algorithm. Such a distribution is essential when a probabilistic representation of backtracking counterfactuals is required. However, in our scenario where $\mathbf{X} = \mathbf{F}(\mathbf{U})$ and $\mathbf{F}(.)$ is invertible, a simpler perspective can be adopted. Here, $\mathbf{U}$ can be treated as a deterministic vector, which simplifies the formulation considerably.

When $\mathbf{U}$ is deterministic, we may treat $\mathbf{U}^{\text{CF}}$ as another deterministic vector close to $\mathbf{U}$, preserving the essence of backtracking without resorting to $\mathbb{P}_B(\mathbf{U}^{\text{CF}} \mid \mathbf{U})$. In interpretability tasks, one typically seeks an input $\mathbf{x}^{\text{CF}}$ near $\mathbf{x}$ that remains faithful to causal constraints. Thus, viewing $\mathbf{x}^{\text{CF}}$ as deterministic naturally aligns with this goal.

Based on this reasoning, we propose our method, BRACE,

with the following optimization problem:

$$\arg \min_{\mathbf{x}^{\text{CF}}, \mathbf{u}^{\text{CF}}} \quad d_X \left( \mathbf{x}, \mathbf{x}^{\text{CF}} \right) + \lambda \, d_U \left( \mathbf{u}, \mathbf{u}^{\text{CF}} \right)$$
$$\text{s.t.} \quad h(\mathbf{x}^{\text{CF}}) = y^{\text{CF}},$$
$$\mathbf{x}^{\text{CF}} = \mathbf{F}(\mathbf{u}^{\text{CF}}), \tag{8}$$
$$\mathbf{x} = \mathbf{F}(\mathbf{u}),$$

which can also be expressed as:

$$\arg \min_{\mathbf{x}^{\text{CF}}} d_X \left( \mathbf{x}, \mathbf{x}^{\text{CF}} \right) + \lambda \, d_U \left( \mathbf{F}^{-1}(\mathbf{x}), \mathbf{F}^{-1}(\mathbf{x}^{\text{CF}}) \right)$$
$$\text{s.t.} \quad h(\mathbf{x}^{\text{CF}}) = y^{\text{CF}}, \tag{9}$$

or equivalently:

$$\arg \min_{\mathbf{u}^{\text{CF}}} \quad d_X \left( \mathbf{x}, \mathbf{F}(\mathbf{u}^{\text{CF}}) \right) + \lambda \, d_U \left( \mathbf{F}^{-1}(\mathbf{x}), \mathbf{u}^{\text{CF}} \right)$$
$$\text{s.t.} \quad h \left( \mathbf{F}(\mathbf{u}^{\text{CF}}) \right) = y^{\text{CF}}. \tag{10}$$

Intuitively, this optimization seeks the closest input $\mathbf{x}^{\text{CF}}$ to $\mathbf{x}$ that achieves the desired output $y^{\text{CF}}$ while preserving the causal relationships encoded in the input variables.

In (8), the objective function includes two terms: $d_X \left( \mathbf{x}, \mathbf{x}^{\text{CF}} \right)$, which ensures the counterfactual input remains close to the observed input, and $d_U \left( \mathbf{u}, \mathbf{u}^{\text{CF}} \right)$, which ensures that the latent variables of the factual and counterfactual worlds are similar. The constraints enforce the desired counterfactual output ($h(\mathbf{x}^{\text{CF}}) = y^{\text{CF}}$), causal consistency ($\mathbf{x}^{\text{CF}} = \mathbf{F}(\mathbf{u}^{\text{CF}})$), and the relationship between the observed input and the latent variables ($\mathbf{x} = \mathbf{F}(\mathbf{u})$).

As $d_U \left( \mathbf{u}, \mathbf{u}^{\text{CF}} \right)$ increases, the counterfactual latent variables $\mathbf{u}^{\text{CF}}$ deviate further from the factual latent variables $\mathbf{u}$, making the counterfactual less connected to the factual observation. The parameter $\lambda$ regulates the trade-off between maintaining proximity in the latent space and ensuring the counterfactual remains close to the original input.

When $\lambda = 0$, the proximity of latent variables is ignored, resulting in solutions that lack causal consistency and focus solely on minimizing the distance between $\mathbf{x}$ and $\mathbf{x}^{\text{CF}}$. Conversely, as $\lambda \to \infty$, the optimization prioritizes minimizing $d_U \left( \mathbf{u}, \mathbf{u}^{\text{CF}} \right)$, which ensures minimal deviation in the latent space but disregards proximity in the input space. The ideal solution balances these objectives, ensuring that the counterfactual is both causally consistent and close to the original input.

## 5. Relation Between Backtracking and Interventional Counterfactuals

Our causal model is represented as $\mathbf{X} = \mathbf{F}(\mathbf{U})$, where $\mathbf{F}(.)$ is an invertible function. Consequently, the distribution of the noise variables conditioned on $\mathbf{X} = \mathbf{x}$ becomes

deterministic. Specifically, all the probability mass of the posterior distribution $\mathbb{P}_C(\mathbf{U} \mid \mathbf{X} = \mathbf{x})$ is concentrated at $\mathbf{u} = \mathbf{F}^{-1}(\mathbf{x})$.

As outlined in Section 2.2, backtracking counterfactuals aim to produce a desired counterfactual outcome by keeping the causal graph unchanged while minimally modifying the noise variables $\mathbf{U}$ after observing $\mathbf{x}$. Although backtracking counterfactuals are typically defined using the backtracking conditional distribution $\mathbb{P}_B(\mathbf{U}^{\mathrm{CF}} \mid \mathbf{U})$, when $\mathbf{F}(.)$ is invertible, the deterministic nature of $\mathbf{U}$ eliminates the necessity for statistical modeling. Instead, we directly analyze the connection between backtracking and interventional counterfactuals via their respective causal equations.

**Theorem 5.1.** *In structural causal models that adhere to the Bijective Generation Mechanism (i.e., $\mathbf{F}(.)$ is invertible), backtracking counterfactuals generalize interventional counterfactuals. Specifically, one of the solutions derived from the backtracking counterfactual formulation always coincides with the interventional counterfactual.*

*Proof.* Consider a counterfactual query involving a subset of variables $\mathbf{X}_{\mathcal{A}}^{\mathrm{CF}} = \mathbf{x}_{\mathcal{A}}^*$. Under the Bijective Generation Mechanism, the posterior distribution $\mathbb{P}_C(\mathbf{U} \mid \mathbf{X} = \mathbf{x})$ assigns probability one to $\mathbf{u} = \mathbf{F}^{-1}(\mathbf{x})$, making $\mathbf{u}$ deterministic. The interventional counterfactuals for this query are defined by the following system of equations:

$$\begin{cases} x_i^{\mathrm{ICF}} = f_i(\mathbf{x}_{\mathrm{pa}(i)}^{\mathrm{ICF}}, u_i), & \forall i \notin \mathcal{A}, \\ x_i^{\mathrm{ICF}} = x_i^*, & \forall i \in \mathcal{A}. \end{cases} \quad (11)$$

In contrast, the backtracking counterfactuals are determined by:

$$\begin{cases} x_i^{\mathrm{BCF}} = f_i(\mathbf{x}_{\mathrm{pa}(i)}^{\mathrm{BCF}}, u_i^{\mathrm{BCF}}), & \forall i \notin \mathcal{A}, \\ x_i^{\mathrm{BCF}} = f_i(\mathbf{x}_{\mathrm{pa}(i)}^{\mathrm{BCF}}, u_i^{\mathrm{BCF}}) = x_i^*, & \forall i \in \mathcal{A}. \end{cases} \quad (12)$$

The key difference between (11) and (12) lies in the adjustment mechanism. Interventional counterfactuals modify the causal graph to enforce $\mathbf{X}_{\mathcal{A}}^{\mathrm{CF}} = \mathbf{x}_{\mathcal{A}}^*$, whereas backtracking counterfactuals achieve the same result by adjusting the noise variables.

Due to the DAG assumption in the causal graph, it is clear that equation (11) has a unique solution. After performing interventions on the set $\mathcal{A}$, we obtain multiple DAGs from which we can derive the unique solution for $\mathbf{x}^{\mathrm{ICF}}$ by starting from the source nodes.

Given that $\mathbf{F}(.)$ is invertible, we define

$$\mathbf{u}_{\mathcal{A}}^{\mathrm{ICF}} = \mathbf{F}^{-1}(\mathbf{x}^{\mathrm{ICF}}). \quad (13)$$

We can also rewrite equation (12) as $\mathbf{x}^{\mathrm{BCF}} = \mathbf{F}(\mathbf{u}^{\mathrm{BCF}})$. By definition, if we substitute $\mathbf{u}^{\mathrm{BCF}} = \mathbf{u}_{\mathcal{A}}^{\mathrm{ICF}}$ into the backtracking equations (12), we arrive at the same counterfactual

solution, $\mathbf{x}^{\mathrm{ICF}}$. Therefore, interventional counterfactuals can be considered a specific case of backtracking counterfactuals.

This reasoning can be generalized to any subset of variables $\mathcal{A}$. For any counterfactual query involving $\mathcal{A}$, we can construct $\mathbf{u}^{\mathrm{ICF}}$ in such a way that backtracking counterfactuals align with interventional counterfactuals. $\qquad \square$

To the best of our knowledge, Theorem 5.1 is the first result that relates backtracking and interventional counterfactuals, and it holds independent significance. Theorem 5.1 demonstrates that when $\mathbf{F}(.)$ is invertible, backtracking counterfactuals inherently include interventional counterfactuals as a specific case. Furthermore, interventional counterfactuals, typically expressed as $\mathbf{x}^{\mathrm{ICF}} = \mathbf{F}_{\mathcal{A}}(\mathbf{u})$, where $\mathbf{F}_{\mathcal{A}}(.)$ represents the structural equations post-intervention, can equivalently be reformulated as $\mathbf{x}^{\mathrm{ICF}} = \mathbf{F}(\mathbf{u}_{\mathcal{A}}^{\mathrm{ICF}})$, bridging the gap between the two paradigms. By the construction (13) in Theorem 5.1, we can see $\mathbf{u}_{\mathcal{A}}^{\mathrm{ICF}} = \mathbf{F}^{-1}\left(\mathbf{F}_{\mathcal{A}}\left(\mathbf{F}^{-1}(\mathbf{x})\right)\right)$.

## 6. Connection Between Our Method and Previous Approaches

Our method BRACE unifies other existing methods in certain scenarios. In our optimization problem (8), setting $\lambda = 0$ simplifies the problem to *Counterfactual Explanations* (4), where causal relationships are disregarded, and the objective becomes finding $\mathbf{x}^{\mathrm{CF}}$ that is closest to $\mathbf{x}$ while modifying the model output.

When $\lambda \to \infty$, (8) reduces to *Deep Backtracking Explanations* (7), which exclusively focuses on finding $\mathbf{u}^{\mathrm{CF}}$ closest to $\mathbf{u}$ without considering the proximity between $\mathbf{x}^{\mathrm{CF}}$ and $\mathbf{x}$, while ensuring the model's output changes.

Our solution (8) can also be interpreted as a special case of *Backtracking Counterfactual Explanations* (6). Specifically, it can be shown that employing the backtracking conditional distribution:

$$\begin{aligned} \mathbb{P}_B(\mathbf{u}^{\mathrm{CF}} \mid \mathbf{u}) \propto \exp\Big( &- d_X\left(\mathbf{F}(\mathbf{u}), \mathbf{F}(\mathbf{u}^{\mathrm{CF}})\right) \\ &- \lambda \cdot d_U\left(\mathbf{u}, \mathbf{u}^{\mathrm{CF}}\right) \Big) \end{aligned} \quad (14)$$

renders (8) equivalent to *Backtracking Counterfactual Explanations* (6). Detailed derivations are provided in the Appendix B, leveraging the theoretical framework from (Von Kügelgen et al., 2023).

While the connections to *Counterfactual Explanations*, *Deep Backtracking Explanations*, and *Backtracking Counterfactual Explanations* are established, a significant question remains: how does our solution (8) relate to *Causal Algorithmic Recourse* (5)? The following theorem provides an answer.

**Theorem 6.1.** *Assume that the distance functions $d_X(\cdot, \mathbf{x})$ and $d_U(\cdot, \mathbf{u})$ are convex, $\mathbf{F}(\cdot)$ and $h(\cdot)$ are linear functions, and the cost function is given as $\mathrm{cost}(\mathcal{A}; \mathbf{x}) = d_X(\mathbf{x}^{\mathrm{CF}}, \mathbf{x})$. Then, our method* BRACE *outperforms Causal Algorithmic Recourse. Specifically, for a fixed distance $\alpha$ in the latent space $d_U(\mathbf{u}^{\mathrm{CF}}, \mathbf{u}) = \alpha$, there exists a $\lambda$ such that the solution of (8) yields a counterfactual $\mathbf{x}^{\mathrm{CF}}$ closer to the observed input $\mathbf{x}$ than the solution of (5).*

*Proof.* We start by reformulating *Causal Algorithmic Recourse* (5) into a form analogous to our proposed solution (8). Using Theorem 5.1, *Causal Algorithmic Recourse* (5) can be rewritten as:

$$\arg\min_{\mathbf{x}^{\mathrm{CF}}, \mathcal{A}} \quad d_X\left(\mathbf{x}^{\mathrm{CF}}, \mathbf{x}\right)$$
$$\begin{aligned}
\text{s.t.} \quad & h(\mathbf{x}^{\mathrm{CF}}) = y^{\mathrm{CF}}, \\
& \mathbf{x}^{\mathrm{CF}} = \mathbf{F}\left(\mathbf{u}_{\mathcal{A}}^{\mathrm{ICF}}\right), \\
& \mathbf{u}_{\mathcal{A}}^{\mathrm{ICF}} = \mathbf{F}^{-1}\left(\mathbf{F}_{\mathcal{A}}(\mathbf{u})\right), \\
& \mathbf{x} = \mathbf{F}(\mathbf{u}).
\end{aligned} \tag{15}$$

The main question is whether there exists a $\lambda$ such that the optimal $\mathbf{x}^{\mathrm{CF}}$ in our method (8) coincides with the optimal $\mathbf{x}^{\mathrm{CF}}$ in (15). Suppose the optimal solution to (15) is attained for $\mathbf{u}_{\mathcal{A}^*}^{\mathrm{ICF}}$. Let $\alpha$ represent the distance between $\mathbf{u}_{\mathcal{A}^*}^{\mathrm{ICF}}$ and $\mathbf{u}$:

$$d_U\left(\mathbf{u}_{\mathcal{A}^*}^{\mathrm{ICF}}, \mathbf{u}\right) = \alpha. \tag{16}$$

We now define the following optimization problem:

$$\arg\min_{\mathbf{x}^{\mathrm{CF}}, \mathbf{u}^{\mathrm{CF}}} \quad d_X\left(\mathbf{x}^{\mathrm{CF}}, \mathbf{x}\right)$$
$$\begin{aligned}
\text{s.t.} \quad & h(\mathbf{x}^{\mathrm{CF}}) = y^{\mathrm{CF}}, \\
& \mathbf{x}^{\mathrm{CF}} = \mathbf{F}\left(\mathbf{u}^{\mathrm{CF}}\right), \\
& d_U\left(\mathbf{u}^{\mathrm{CF}}, \mathbf{u}\right) = \alpha, \\
& \mathbf{x} = \mathbf{F}(\mathbf{u}).
\end{aligned} \tag{17}$$

Let the optimal solution to (15) be $\mathbf{x}^{*\mathrm{ICF}}$, and the optimal solution to (17) be $\mathbf{x}^{*\mathrm{BCF}}$. Then, it follows:

$$d_X\left(\mathbf{x}^{*\mathrm{BCF}}, \mathbf{x}\right) \le d_X\left(\mathbf{x}^{*\mathrm{ICF}}, \mathbf{x}\right). \tag{18}$$

This inequality holds because $\mathbf{u}_{\mathcal{A}^*}^{\mathrm{ICF}}$ satisfies the constraint $d_U\left(\mathbf{u}^{\mathrm{CF}}, \mathbf{u}\right) = \alpha$, while other feasible values of $\mathbf{u}^{\mathrm{CF}}$ within the same constraint may reduce the objective $d_X\left(\mathbf{x}^{\mathrm{CF}}, \mathbf{x}\right)$ further. Hence, the *Causal Algorithmic Recourse* formulation (15) may not always yield the closest $\mathbf{x}^{\mathrm{CF}}$ to $\mathbf{x}$ among all $\mathbf{u}^{\mathrm{CF}}$ satisfying the distance constraint $\alpha$ from $\mathbf{u}$.

Next, we examine whether there exists a $\lambda$ such that the optimal solution of our method (8) aligns with the optimal solution of (17). In essence, we seek a $\lambda$ such that

the optimal $\mathbf{u}^{\mathrm{CF}}$ from (8) satisfies the distance constraint $d_U\left(\mathbf{u}^{\mathrm{CF}}, \mathbf{u}\right) = \alpha$.

To approach this, consider the following vector optimization problem:

$$\arg\min_{\mathbf{x}^{\mathrm{CF}}, \mathbf{u}^{\mathrm{CF}}} \quad \left(d_X\left(\mathbf{x}, \mathbf{x}^{\mathrm{CF}}\right), \ d_U\left(\mathbf{u}, \mathbf{u}^{\mathrm{CF}}\right)\right)$$
$$\begin{aligned}
\text{s.t.} \quad & h(\mathbf{x}^{\mathrm{CF}}) = y^{\mathrm{CF}}, \\
& \mathbf{x}^{\mathrm{CF}} = \mathbf{F}(\mathbf{u}^{\mathrm{CF}}), \\
& \mathbf{x} = \mathbf{F}(\mathbf{u}).
\end{aligned} \tag{19}$$

The optimization (19) simultaneously minimizes $d_X\left(\mathbf{x}, \mathbf{x}^{\mathrm{CF}}\right)$ and $d_U\left(\mathbf{u}, \mathbf{u}^{\mathrm{CF}}\right)$. However, in certain cases, reducing one term may result in an increase in the other.

To resolve this, we utilize the concept of Pareto optimality (Boyd & Vandenberghe, 2004). A well-known result for convex problems is that scalarizing the objective:

$$d_X\left(\mathbf{x}, \mathbf{x}^{\mathrm{CF}}\right) + \lambda \, d_U\left(\mathbf{u}, \mathbf{u}^{\mathrm{CF}}\right) \tag{20}$$

yields all Pareto-optimal solutions by varying $\lambda > 0$. Specifically, every optimal solution of the scalarized optimization corresponds to a Pareto-optimal point of the vector optimization. Moreover, since the vector optimization problem (19) is convex (from the assumptions of the theorem), all Pareto-optimal points can be achieved.

Returning to our optimization, note that the solution to (17) is a Pareto-optimal point of (19) because with constraint $d_U\left(\mathbf{u}^{\mathrm{CF}}, \mathbf{u}\right) = \alpha$ we minimizes $d_X\left(\mathbf{x}^{\mathrm{CF}}, \mathbf{x}\right)$. Thus, the solution cannot be further improved along the $d_X\left(\mathbf{x}^{\mathrm{CF}}, \mathbf{x}\right)$ axis.

Thus, by varying $\lambda$, it is possible to identify a $\lambda$ such that the solution of our method BRACE (8) matches the solution of (17), ensuring $d_U\left(\mathbf{u}^{\mathrm{CF}}, \mathbf{u}\right) = \alpha$. Consequently, as demonstrated in (18), our proposed method yields a $\mathbf{x}^{\mathrm{CF}}$ that is closer to $\mathbf{x}$ compared to *Causal Algorithmic Recourse*, while preserving the fixed distance $\alpha$ between the latent variables $\mathbf{u}$ and $\mathbf{u}^{\mathrm{CF}}$. $\qquad \square$

The convexity of the distance functions, along with the linearity of $\mathbf{F}(\cdot)$ and $h(\cdot)$, are assumed primarily to ensure the existence of $\lambda$ by utilizing the convexity of the vector optimization problem (19). However, similar conclusions can still be derived without these assumptions if there exists a suitable $\lambda$ such that $d_U\left(\mathbf{u}^{\mathrm{CF}}, \mathbf{u}\right) = \alpha$. This indicates that, even in cases where the convexity of $d_X(\cdot, \mathbf{x})$, $d_U(\cdot, \mathbf{u})$, or the linearity of $\mathbf{F}(\cdot)$ and $h(\cdot)$ are not assumed, our proposed method often provides a $\mathbf{x}^{\mathrm{CF}}$ that is closer to $\mathbf{x}$ than *Causal Algorithmic Recourse*, while preserving the fixed distance $\alpha$ between the latent variables $\mathbf{u}$ and $\mathbf{u}^{\mathrm{CF}}$.

Importantly, convexity and linearity assumptions are not necessary for the core insight to remain valid. Even without assuming convexity of the distance functions or linearity of $\mathbf{F}(\cdot)$ and $h(\cdot)$, we can *always* find a solution that outperforms *Causal Algorithmic Recourse* among the Pareto optimal points of the vector optimization problem (19). The assumptions of linearity and convexity are only required to guarantee that this Pareto optimal point can be *captured* by some value of $\lambda$.

It is worth noting that our method is substantially more efficient computationally than *Causal Algorithmic Recourse*, since that approach relies on a combinatorial optimization procedure. Moreover, our method also surpasses *Backtracking Counterfactual Explanations* in computational efficiency, especially when dealing with a complex distribution $\mathbb{P}_B$.

# 7. Metric Selection and Optimization Approach

To solve the optimization problem in Eq. (8), it is essential to define the distance metrics $d_X\left(\mathbf{x}, \mathbf{x}^{\mathrm{CF}}\right)$ and $d_U\left(\mathbf{u}, \mathbf{u}^{\mathrm{CF}}\right)$. For $d_X(.,.)$, which measures proximity in the observed space, the $\ell_1$ norm is a natural choice as it minimizes the number of modified features, making the counterfactuals more interpretable and actionable:

$$d_X\left(\mathbf{x}, \mathbf{x}^{\mathrm{CF}}\right) = \left\|\mathbf{x} - \mathbf{x}^{\mathrm{CF}}\right\|_1. \tag{21}$$

For the latent space, $d_U(.,.)$ evaluates how plausible a counterfactual is relative to the original latent representation. The $\ell_2$ norm ensures smoothness and proximity:

$$d_U\left(\mathbf{u}, \mathbf{u}^{\mathrm{CF}}\right) = \left\|\mathbf{u} - \mathbf{u}^{\mathrm{CF}}\right\|_2. \tag{22}$$

By combining these metrics, the optimization problem can be reformulated in a meaningful way.

Solving the optimization problem in Eq. (8), particularly for complex models such as neural networks (Katz et al., 2017) or additive tree models (Ates et al., 2021), is generally *NP-*hard. Gradient-based methods are effective when both the objective and the constraints are differentiable. For example, the constraints can be integrated into the objective function as penalty terms:

$$\arg\min_{\mathbf{x}^{\mathrm{CF}}} \; d_X(\mathbf{x}, \mathbf{x}^{\mathrm{CF}}) + \lambda \, d_U(\mathbf{F}^{-1}(\mathbf{x}), \mathbf{F}^{-1}(\mathbf{x}^{\mathrm{CF}})) \\ + \beta \, \mathrm{Loss}\left(h(\mathbf{x}^{\mathrm{CF}}), y^{\mathrm{CF}}\right), \tag{23}$$

where $\mathrm{Loss}(.)$ is a common classification loss, such as cross-entropy. To approximate solutions in practice, $\beta$ is gradually increased until the counterfactual $\mathbf{x}^{\mathrm{CF}}$ satisfies the desired output class $y^{\mathrm{CF}}$ (Szegedy et al., 2013).

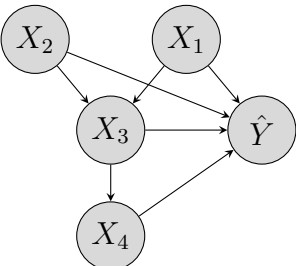

Figure 1. Causal graph of the bank's high-risk detection model. $X_1$ is gender, $X_2$ is age, $X_3$ is loan amount, and $X_4$ is repayment duration in months. The model's output $\hat{Y}$ indicates high or low risk for loan approval.

Heuristic approaches also provide practical alternatives; for instance, shortest path searches in empirical graphs (Poyiadzi et al., 2020) or expanding-sphere searches (Laugel et al., 2017) offer approximate solutions in specific scenarios.

# 8. Experimental Evaluation

## 8.1. Simulation Setup

To evaluate the proposed method, we adopt the experiment in the *Causal Algorithmic Recourse* paper (Karimi et al., 2021), as it serves as a critical baseline for comparison. Since the primary focus is on assessing the interpretability of the proposed method, we use a simple model for $h(\cdot)$. This ensures that the exact solutions to the optimization problem can be computed and aligned with our intuitive understanding of the task.

We consider a model $h(\cdot)$ designed to classify individuals as high- or low-risk for loan approval. The input vector $\mathbf{X}$ is assumed to follow the given causal structure:

$$\begin{aligned} X_1 &:= U_1, \\ X_2 &:= U_2, \\ X_3 &:= f_3(X_1, X_2) + U_3, \\ X_4 &:= f_4(X_3) + U_4, \end{aligned} \tag{24}$$

where the system's output is given by $\hat{Y} = h(X_1, X_2, X_3, X_4)$. Figure 1 illustrates the causal graph associated with the problem.

For this simulation, the causal graph is assumed to be known, while the functions $f_3(\cdot)$, $f_4(\cdot)$, and $h(\cdot)$ are estimated using real-world data from the *German Credit Dataset* (Hofmann, 1994). We assume $f_3(\cdot)$ and $f_4(\cdot)$ are linear functions and $h(\cdot)$ is logistic regression. Following (Peters et al., 2017), the coefficients of the causal model can be derived using linear regression when the causal functions are linear.

The feature $X_1$ (gender) is one-hot encoded during logistic regression for $h(\cdot)$ and is kept fixed when generating coun-

*Table 1.* Counterfactual solutions from different methods for an individual originally classified as high-risk $\big(\mathbf{x} = (\text{female}, 24, \$4308, 48)\big)$. Each method modifies the features to flip the prediction to low-risk.

| Method | Gender | Age | Loan amount | Duration |
|---|---|---|---|---|
| Original (High-risk) | female | 24 | $4308 | 48 |
| BRACE (**Our Method**, $\lambda = 1$) | female | 24 | $4087 | 33.0 |
| BRACE (**Our Method**, $\lambda = 1.2$) | female | 24 | $3736 | 33.3 |
| Counterfactual Explanations (Wachter et al., 2017) | female | 24 | $4308 | 32.8 |
| Causal Algorithmic Recourse (Karimi et al., 2021) | female | 24 | $4308 | 32.8 |
| Deep Backtracking Explanations (Kladny et al., 2024) | female | 27.2 | $2727 | 35.7 |

terfactuals. Since $X_1$ is categorical, modifying it is avoided, as changing gender does not provide actionable insights. Additionally, all features are normalized by their standard deviations to improve performance.

### 8.2. Optimization Problem and Results

We consider an individual with features $\mathbf{x} = (\text{female}, 24, \$4308, 48)$ classified as high-risk by the model $h(\cdot)$. Using the causal model and the learned functions $f_3(\cdot)$ and $f_4(\cdot)$, we derive the latent representation $\mathbf{u}$. The following optimization problem is then formulated:

$$
\underset{\mathbf{x}^{\text{CF}}, \mathbf{u}^{\text{CF}}}{\arg\min} \sum_{i=2}^{4} \frac{|x_i - x_i^{\text{CF}}|}{\sigma_i} + \lambda \sqrt{\sum_{i=2}^{4} \frac{\left(u_i - u_i^{\text{CF}}\right)^2}{\sigma_i^2}}
$$
$$
\begin{aligned}
\text{s.t.} \quad & h(\mathbf{x}^{\text{CF}}) = \text{low-risk}, \\
& x_2^{\text{CF}} = u_2^{\text{CF}}, \\
& x_3^{\text{CF}} = f_3(x_1^{\text{CF}}, x_2^{\text{CF}}) + u_3^{\text{CF}}, \\
& x_4^{\text{CF}} = f_4(x_3^{\text{CF}}) + u_4^{\text{CF}}.
\end{aligned} \tag{25}
$$

We solve the optimization problem in (25) with $\lambda = 1$ and $\lambda = 1.2$. Table 1 reports $\mathbf{x}^{\text{CF}}$ from our method and other prominent approaches for comparison.

As shown in Table 1, both Counterfactual Explanations and Causal Algorithmic Recourse focus solely on reducing the repayment duration, which is not actionable for the user and fails to provide meaningful guidance for future improvements. In contrast, the Deep Backtracking Explanations alters all features, leading to a significant departure from the original observation and reducing local interpretability. Our approach finds a balance by adjusting both the loan amount and repayment duration while maintaining sparsity and interpretability, offering a more intuitive and actionable explanation for the user.

When the user's initial features are given by $x = (\text{female}, 24, \$4308, 48)$, this suggests that a 48-month loan repayment is appropriate for the user. Therefore, if we adjust this feature vector solely by reducing the repayment

duration, the repayment becomes significantly more challenging for the user, thereby making the explanation less actionable.

To put it quantitatively, repaying $4308 over 48 months corresponds to a monthly payment of $89.75. Any explanation that deviates considerably from this monthly rate is less actionable. Counterfactual Explanations and Causal Algorithmic Recourse yield a monthly repayment of about $131.3 (i.e., $4308 divided by 32.8 months). In contrast, our solution results in a monthly repayment of $123.8 when $\lambda = 1$ (i.e., $4087 divided by 33 months) and $112.2 when $\lambda = 1.2$ (i.e., $3736 divided by 33.3 months). Thus, while Counterfactual Explanations and Causal Algorithmic Recourse increase the monthly repayment by 46.3%, our approach leads to increases of 37.8% and 25.0% for $\lambda = 1$ and $\lambda = 1.2$, respectively, making our explanations more actionable for the user.

Table 2 provides the counterfactual outcomes for another example with initial features $x = (\text{male}, 27, \$14027, 60)$, where similar trends across methods are observed.

### 8.3. Sensitivity Analysis

Estimating causal functions in the input graph often involves approximations, introducing potential noise. To evaluate the robustness of our method, we add zero-mean Gaussian noise with a standard deviation of 5 to the coefficients of $f_3(\cdot)$ and solve the optimization problem (25) with $\lambda = 1.2$ for an individual with features $\mathbf{x} = (\text{female}, 24, \$4308, 48)$. The resulting counterfactuals are:

$$
\begin{aligned}
\mathbf{x}^{\text{CF}} &= (\text{female}, 24, \$3839, 33.2), \\
\mathbf{x}^{\text{CF}} &= (\text{female}, 24, \$3572, 33.5), \\
\mathbf{x}^{\text{CF}} &= (\text{female}, 24, \$3618, 33.4).
\end{aligned} \tag{26}
$$

Despite significant noise, the results remain stable, with age unchanged and explanations staying sparse. This highlights the robustness of our method, showing that even approximate causal functions produce reliable and interpretable counterfactuals.

*Table 2.* Counterfactual solutions from different methods for an individual originally classified as high-risk.

| Method | Gender | Age | Loan amount | Duration |
|---|---|---|---|---|
| Original (High-risk) | male | 27 | $14027 | 60 |
| BRACE (**Our Method**, $\lambda = 1$) | male | 27 | $13686 | 36.9 |
| BRACE (**Our Method**, $\lambda = 1.2$) | male | 27 | $13149 | 37.4 |
| Counterfactual Explanations (Wachter et al., 2017) | male | 27 | $14027 | 36.6 |
| Causal Algorithmic Recourse (Karimi et al., 2021) | male | 27 | $14027 | 36.6 |
| Deep Backtracking Explanations (Kladny et al., 2024) | male | 31.9 | $11599 | 41.1 |

## 9. Conclusion

In this work, we presented a new framework BRACE for counterfactual explanations based on backtracking counterfactuals. Our approach overcomes the limitations of interventional counterfactuals by introducing an optimization problem that generates actionable and causally consistent explanations. By solving a single unified objective parameterized by $\lambda$, BRACE recovers four established paradigms—classical *Counterfactual Explanations* ($\lambda = 0$), *Deep Backtracking Explanations* ($\lambda \to \infty$), *Backtracking Counterfactual Explanations* via a specific backtracking conditional distribution, and *Causal Algorithmic Recourse* under a convexity assumption. Additionally, we demonstrated that our method is both easier to understand and more computationally efficient compared to causal algorithmic recourse. Through simulation experiments, we verified that the proposed method produces explanations that are more intuitive for users and more practical for real-world applications.

## 10. Future Work

This work opens several directions for further research:

*Relaxing Assumptions for Connection with Causal Algorithmic Recourse*: Our approach depends on convexity assumptions to establish a connection with causal algorithmic recourse. Future work could investigate alternative conditions that do not require these assumptions, allowing the theory to be applied to non-linear and non-convex models frequently found in real-world applications.

*Testing on Complex Models*: This study focused on simpler models for $h(.)$ to ensure intuitive understanding of the task. A valuable next step is to test our method on more complex models, such as deep neural networks, and compare its performance with existing state-of-the-art methods. This would help demonstrate the method's effectiveness in handling challenging real-world scenarios.

*Improving Backtracking Counterfactual Definitions*: The current definition of backtracking counterfactuals does not

ensure that the noise variables $\mathbf{U}^{\text{CF}}$ in the counterfactual world remain mutually independent. In SCMs, this independence is important for maintaining the causal interpretation of the noise variables. Extending the definition to enforce this independence would enhance both the **theoretical consistency** and **practical usefulness** of backtracking counterfactuals, making them more aligned with core principles of causal reasoning.

## Acknowledgements

This research was conducted while Pouria Fatemi and Ehsan Sharifian were affiliated with Sharif University of Technology.

## Impact Statement

Our work advances the field of machine learning by proposing a framework for counterfactual explanations that enhances interpretability while ensuring causal consistency and actionability. This method unifies multiple existing approaches, including counterfactual explanations, deep backtracking explanations, causal algorithmic recourse, and backtracking counterfactual explanations, while improving computational efficiency.

The potential impact of our work is most relevant to high-stakes applications such as finance and healthcare, where reliable and transparent decision-making is essential. By incorporating causal reasoning into counterfactual explanations, our approach contributes to making AI-driven decisions more interpretable and aligned with real-world constraints. While our method does not introduce direct ethical concerns, its application in automated decision-making should be carefully evaluated to ensure fair and responsible use.

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

# A. Formal Definition of Interventional and Backtracking Counterfactuals

To formally understand the differences and computation processes underlying interventional and backtracking counterfactuals, we outline their respective definitions and procedural steps below.

## A.1. Interventional Counterfactuals

1. **Abduction**: Update the distribution of the noise variables $\mathbf{U}$ in the causal model from $P_{\mathbf{U}}$ to the posterior distribution $P_{\mathbf{U}|\mathbf{X}=\mathbf{x}}$, using the observed factual data $\mathbf{x}$.

2. **Action**: Perform a hard intervention $do(X_i := x_i^{\mathrm{CF}})$ for $i \in \mathcal{A}$, modifying the structural equations of the causal model. Denote the modified structural equations as $\mathbf{S}^{\mathrm{CF}}$, while retaining the original equations $f_i^{\mathrm{CF}} = f_i$ for $i \notin \mathcal{A}$.

3. **Prediction**: Using the updated causal model $\mathcal{C}^{\mathrm{CF}} = (\mathbf{S}^{\mathrm{CF}}, P_{\mathbf{U}|\mathbf{X}=\mathbf{x}})$, compute the distribution over the desired counterfactual outcomes $\mathbf{Y}^{\mathrm{CF}}$.

## A.2. Backtracking Counterfactuals

1. **Cross-World Abduction**: Update the joint distribution $\mathbb{P}_B(\mathbf{U}^{\mathrm{CF}}, \mathbf{U}) = \mathbb{P}(\mathbf{U}) \mathbb{P}_B(\mathbf{U}^{\mathrm{CF}} \mid \mathbf{U})$ using the variables $(\mathbf{x}_{\mathcal{A}}^{\mathrm{CF}}, \mathbf{x})$ to obtain the posterior distribution $\mathbb{P}_B(\mathbf{U}^{\mathrm{CF}}, \mathbf{U} \mid \mathbf{x}_{\mathcal{A}}^{\mathrm{CF}}, \mathbf{x})$:

$$\mathbb{P}_B(\mathbf{u}^{\mathrm{CF}}, \mathbf{u} \mid \mathbf{x}_{\mathcal{A}}^{\mathrm{CF}}, \mathbf{x}) = \frac{\mathbb{P}_B(\mathbf{u}^{\mathrm{CF}}, \mathbf{u}) \, 1\{\mathbf{F}_{\mathcal{A}}(\mathbf{u}^{\mathrm{CF}}) = \mathbf{x}_{\mathcal{A}}^{\mathrm{CF}}\} \, 1\{\mathbf{F}(\mathbf{u}) = \mathbf{x}\}}{\mathbb{P}_B(\mathbf{x}_{\mathcal{A}}^{\mathrm{CF}}, \mathbf{x})}. \tag{27}$$

Where

$$\mathbb{P}_B(\mathbf{x}_{\mathcal{A}}^{\mathrm{CF}}, \mathbf{x}) = \int \mathbb{P}_B(\mathbf{u}^{\mathrm{CF}}, \mathbf{u}) \, 1\{\mathbf{F}_{\mathcal{A}}(\mathbf{u}^{\mathrm{CF}}) = \mathbf{x}_{\mathcal{A}}^{\mathrm{CF}}\} \, 1\{\mathbf{F}(\mathbf{u}) = \mathbf{x}\} \, d\mathbf{u} \, d\mathbf{u}^{\mathrm{CF}}. \tag{28}$$

Calculating $\mathbb{P}_B(\mathbf{x}_{\mathcal{A}}^{\mathrm{CF}}, \mathbf{x})$ becomes computationally challenging for complex $\mathbb{P}_B(\mathbf{U}^{\mathrm{CF}}, \mathbf{U})$ distributions, as we must integrate over all values of this distribution.

2. **Marginalization**: Marginalize over $\mathbf{U}$ to compute the posterior distribution $\mathbb{P}_B(\mathbf{U}^{\mathrm{CF}} \mid \mathbf{x}_{\mathcal{A}}^{\mathrm{CF}}, \mathbf{x})$:

$$\mathbb{P}_B(\mathbf{u}^{\mathrm{CF}} \mid \mathbf{x}_{\mathcal{A}}^{\mathrm{CF}}, \mathbf{x}) = \int \mathbb{P}_B(\mathbf{u}^{\mathrm{CF}}, \mathbf{u} \mid \mathbf{x}_{\mathcal{A}}^{\mathrm{CF}}, \mathbf{x}) \, d\mathbf{u}. \tag{29}$$

3. **Prediction**: Using the updated causal graph with noise distribution $\mathbb{P}_B(\mathbf{U}^{\mathrm{CF}} \mid \mathbf{x}_{\mathcal{A}}^{\mathrm{CF}}, \mathbf{x})$, compute the probability of the desired counterfactual event:

$$\mathbb{P}_B(\mathbf{y}^{\mathrm{CF}} \mid \mathbf{x}_{\mathcal{A}}^{\mathrm{CF}}, \mathbf{x}) = \int \mathbb{P}_B(\mathbf{u}^{\mathrm{CF}} \mid \mathbf{x}_{\mathcal{A}}^{\mathrm{CF}}, \mathbf{x}) \, 1\{\mathbf{F}(\mathbf{u}^{\mathrm{CF}}) = \mathbf{y}^{\mathrm{CF}}\} \, d\mathbf{u}^{\mathrm{CF}}. \tag{30}$$

# B. Relation of Our Solution to Backtracking Counterfactual Explanations

We aim to demonstrate that our solution (8) can be connected to the backtracking counterfactual explanations framework presented in (Von Kügelgen et al., 2023), which is formulated as the optimization problem (6), by considering a specific choice of $\mathbb{P}_B(\mathbf{U}^{\mathrm{CF}} \mid \mathbf{U})$. This connection is established by following the three steps of backtracking counterfactual computation:

1. **Cross-World Abduction**: Compute the posterior distribution of the latent variables in the causal graph. Given that the function $\mathbf{F}(.)$ is invertible, we have:

$$\mathbb{P}_B\left(\mathbf{u}^{\mathrm{CF}}, \mathbf{u} \mid y^{\mathrm{CF}}, \mathbf{x}\right) = \mathbb{P}_B\left(\mathbf{u}^{\mathrm{CF}} \mid \mathbf{u}, y^{\mathrm{CF}}, \mathbf{x}\right) \mathbb{P}_B\left(\mathbf{u} \mid y^{\mathrm{CF}}, \mathbf{x}\right) \tag{31}$$

$$= \mathbb{P}_B\left(\mathbf{u}^{\mathrm{CF}} \mid \mathbf{u}, y^{\mathrm{CF}}\right) \mathbb{P}_B(\mathbf{u} \mid \mathbf{x}) \tag{32}$$

$$= \mathbb{P}_B\left(\mathbf{u}^{\mathrm{CF}} \mid \mathbf{u}, y^{\mathrm{CF}}\right) 1\left\{\mathbf{F}^{-1}(\mathbf{x}) = \mathbf{u}\right\}. \tag{33}$$

2. **Marginalization**: Compute the marginal posterior distribution over $\mathbf{u}^{\mathrm{CF}}$. Since the entire probability mass is concentrated at the point $\mathbf{u} = \mathbf{F}^{-1}(\mathbf{x})$, we have:

$$\mathbb{P}_B\left(\mathbf{u}^{\mathrm{CF}} \mid y^{\mathrm{CF}}, \mathbf{x}\right) = \mathbb{P}_B\left(\mathbf{u}^{\mathrm{CF}}, \mathbf{u} = \mathbf{F}^{-1}(\mathbf{x}) \mid y^{\mathrm{CF}}, \mathbf{x}\right) \tag{34}$$

$$= \mathbb{P}_B\left(\mathbf{u}^{\mathrm{CF}} \mid \mathbf{F}^{-1}(\mathbf{x}), y^{\mathrm{CF}}\right). \tag{35}$$

3. **Prediction**: Compute the distribution $\mathbf{X}^{\mathrm{CF}} \mid y^{\mathrm{CF}}, \mathbf{x}$. Since $\mathbf{F}(.)$ is deterministic, we have:

$$\mathbf{u}^{\mathrm{CF}} \sim \mathbf{U}^{\mathrm{CF}} \mid \mathbf{F}^{-1}(\mathbf{x}), y^{\mathrm{CF}}, \qquad \mathbf{x}^{\mathrm{CF}} = \mathbf{F}(\mathbf{u}^{\mathrm{CF}}). \tag{36}$$

Now, consider a specific choice for the backtracking conditional distribution:

$$\mathbb{P}_B(\mathbf{u}^{\mathrm{CF}} \mid \mathbf{u}) \propto \exp\left\{-d_X\left(\mathbf{F}(\mathbf{u}), \mathbf{F}(\mathbf{u}^{\mathrm{CF}})\right) - \lambda\, d_U\left(\mathbf{u}, \mathbf{u}^{\mathrm{CF}}\right)\right\}. \tag{37}$$

Substituting this into the posterior distribution of $\mathbf{U}^{\mathrm{CF}} \mid \mathbf{F}^{-1}(\mathbf{x}), y^{\mathrm{CF}}$, we obtain:

$$\mathbf{U}^{\mathrm{CF}} \mid \mathbf{F}^{-1}(\mathbf{x}), y^{\mathrm{CF}} \propto \begin{cases} \exp\left\{-d_X\left(\mathbf{x}, \mathbf{F}(\mathbf{u}^{\mathrm{CF}})\right) - \lambda\, d_U\left(\mathbf{F}^{-1}(\mathbf{x}), \mathbf{u}^{\mathrm{CF}}\right)\right\}, & \text{if } h(\mathbf{x}^{\mathrm{CF}}) = y^{\mathrm{CF}}, \\ 0, & \text{otherwise.} \end{cases} \tag{38}$$

Taking the logarithm on both sides, we have:

$$\log \mathbb{P}_B\left(\mathbf{u}^{\mathrm{CF}} \mid \mathbf{F}^{-1}(\mathbf{x}), y^{\mathrm{CF}}\right) \propto \begin{cases} -d_X\left(\mathbf{x}, \mathbf{F}(\mathbf{u}^{\mathrm{CF}})\right) - \lambda\, d_U\left(\mathbf{F}^{-1}(\mathbf{x}), \mathbf{u}^{\mathrm{CF}}\right), & \text{if } h(\mathbf{x}^{\mathrm{CF}}) = y^{\mathrm{CF}}, \\ -\infty, & \text{otherwise.} \end{cases} \tag{39}$$

Thus, we have:

$$\arg\max_{\mathbf{u}^{\mathrm{CF}}} \log \mathbb{P}_B\left(\mathbf{u}^{\mathrm{CF}} \mid \mathbf{F}^{-1}(\mathbf{x}), y^{\mathrm{CF}}\right) \equiv \arg\min_{\mathbf{u}^{\mathrm{CF}}} \quad d_X\left(\mathbf{x}, \mathbf{F}(\mathbf{u}^{\mathrm{CF}})\right) + \lambda\, d_U\left(\mathbf{F}^{-1}(\mathbf{x}), \mathbf{u}^{\mathrm{CF}}\right),$$

$$\text{s.t.} \quad h(\mathbf{x}^{\mathrm{CF}}) = y^{\mathrm{CF}}. \tag{40}$$

As shown in (40), the optimization problem (10) aligns with backtracking counterfactual explanations (6). Therefore, our solution provides a valid interpretation based on backtracking counterfactuals.

