# OpenReview forum: "A New Approach to Backtracking Counterfactual Explanations: A Unified Causal Framework for Efficient Model Interpretability"
_ICML.cc/2025/Conference — ICML 2025 poster_

### Official Review · Reviewer_Fw7M · 2025-03-12

**Overall Recommendation:** 1

**Summary:**

The paper presents an unifying view on generating counterfactual explanations via backtracking. Namely, the authors propose an optimization objective integrating insights from causal algorithmic recourse and backtracking counterfactual explanations. The paper shows how this new objective subsumes the previous definitions in particular cases. Then, the approach is tested on a synthetic causal setting from the literature.

**Claims And Evidence:**

The authors do not provide a sound justification why the proposed objective (Equation 8) improves from the limitations of (deep) backtracking counterfactuals and related methods. Namely, the authors dismiss too easily the backtracking distribution $P_B(\mathbf{U}^{CF} \mid \mathbf{U})$. Indeed, in Equation 8, optimizing over the distance term over the exogenous variables $d(\mathbf{u}, \mathbf{u}^{CF})$ does not ensure we are finding reasonable changes to the exogenous variables. Thus, falling in the same issue as Equation (4).

Theorem 6.1 holds under very specific conditions (e.g., linearity of the classifier and convexity of the distance functions). It is not clear to me how the theorem is still valid when those conditions are not satisfied (line 294-305 after the proof).

It is not clear how optimizing Equation (24) should improve over the difficulties highlighted by causal algorithmic recourse (Section 5). Indeed, even if we are employing a gradient based approach, we still need to specify a feasible set of features to work on $\mathcal{A}$ (as they also do in [1]).

[1] Dominguez-Olmedo, Ricardo, Amir H. Karimi, and Bernhard Schölkopf. "On the adversarial robustness of causal algorithmic recourse." International Conference on Machine Learning. PMLR, 2022.

**Essential References Not Discussed:**

None.

**Experimental Designs Or Analyses:**

See “Methods And Evaluation Criteria”.

**Methods And Evaluation Criteria:**

Since the paper provides a new objective function, I believe a robust empirical evaluation is needed to better understand the practical implications of the proposed approach.

The experiments are very limited, and they do not provide any statistical guarantees or proper evaluation of optimizing Equation 24 with respect to the alternatives. Namely, the authors focus on a single simulated scenario, by showing us only the results of a unique instance. Moreover, it would be nice to see empirical evaluations of conclusions derived from the theorems (Theorem 6), which are also lacking.

I would suggest the authors to increase the sample size, provide analytical evaluation metrics (e.g., validity, sparsity, distance, etc.), and test the approach over multiple (even synthetics) causal datasets. Further ablations could consider linear/non-linear ANMs (as done by Karimi [1]), more complex classifiers (as done by Dominguez [3]), and approximate SCMs fitted from the data. The assumption of knowing the full SCM does not hold in practice, and optimizing Equation 24 with an approximate SCM would provide interesting insights.

[1] (Adult and COMPAS) Razieh Nabi and Ilya Shpitser. Fair inference on outcomes. In Proceedings of the AAAI Conference on Artificial Intelligence, volume 32, 2018.

[2] (Loan) Karimi et al., Algorithmic recourse under imperfect causal knowledge: a probabilistic approach. Advances in Neural Information Processing Systems, 33:265–277, 2020.

[3] (Further datasets) Dominguez-Olmedo et al., "On the adversarial robustness of causal algorithmic recourse." International Conference on Machine Learning. PMLR, 2022.

**Other Comments Or Suggestions:**

None.

**Other Strengths And Weaknesses:**

None.

**Questions For Authors:**

- Can you better clarify the implications of the various assumptions made in Theorem 6.1 when they do not hold?
- Can you better clarify the potential challenges arising when optimizing Equation 24 with respect to classical differentiable recourse?

**Relation To Broader Scientific Literature:**

The paper is surely interesting for the broad field of Explainable AI.

**Theoretical Claims:**

I checked the claims in Theorem 5.1 and Theorem 6.1 and they seem to be correct (although Theorem 6.1 does require strong assumptions).

---

> ### Author Rebuttal · Authors · 2025-04-01
>
> Thank you for your feedback on our paper. We look forward to exploring your suggestions further to enhance our work.
>
> We do not dismiss the backtracking distribution. As stated in Equation (15) of our paper, our method can be viewed as a special case of Backtracking Counterfactual Explanations by using the backtracking conditional distribution presented there.
>
> Regarding your questions about our experiments, please refer to our **Answer for Q1, Answer for Q3, and Answer for Q4** in our rebuttal to Reviewer 7Lfk.
>
> Concerning your question about generalizing our problem to cases where the input SCM is not fully known or is non-invertible: suppose that $\mathbf{F}(\cdot)$ is non-invertible and not fully known. In this scenario, after observing $x$, the posterior over $U$ is no longer a point mass. Instead, we can compute the distribution $U \mid x$ using the available causal relationships and the prior $P(U)$. An interesting scenario arises when the causal graph among variables is known, yet the specific functional forms are not. Following the approach in [1], we can assume a probability distribution (e.g., a Gaussian process) over each causal function. By combining this with the prior on $U$, we can compute the posterior distribution $U \mid x$. In this case, both $\mathbf{F}(\cdot)$ and $U \mid x$ become random, leading to the following optimization:
>
> \begin{split}
> \arg\min_{u^{\mathrm{CF}}} \quad & E_{F} \left[d_X\left(x,F(u^{\mathrm{CF}})\right)\right] + \lambda \ E_{U \mid x}\left[d_U\left(U, u^{\mathrm{CF}} \right)\right] \\
> \quad \text{s.t.} \quad & E_{F} \left[h\left(F(u^{\mathrm{CF}})\right)\right] \ge \alpha
> \end{split}
>
> In practice, heuristic methods are used to solve these optimization problems, as discussed in [1].
>
> > Can you better clarify the implications of the various assumptions made in Theorem 6.1 when they do not hold?
>
> It should be mentioned that, to the best of our knowledge, Theorem 5.1 in our paper is the first result that relates backtracking and interventional counterfactuals, and it holds independent significance. Additionally, Theorem 6.1 is, as far as we know, the first result that connects causal algorithmic recourse with backtracking explanation methods.
>
> Assumptions of linearity and convexity in our models are necessary because we require the vector optimization (Equation 20) to be convex. This convexity ensures that by varying $\lambda$, we can capture all Pareto optimal solutions, thereby guaranteeing that we also obtain the desired Pareto optimal solution in Equation (18). However, as mentioned in the paper (lines 295–305), even without any additional assumptions—relying solely on the ANM assumption—we can *always* find a better solution than causal algorithmic recourse among the Pareto optimal points of the vector optimization (Equation 20). The assumptions of linearity and convexity ensure that we can *capture* this Pareto optimal point for some value of $\lambda$. That said, many Pareto optimal points of a non-convex vector optimization can still be reached, and our required Pareto optimal solution (the solution of Equation (18)) might be one of them. In essence, linearity and convexity assumptions ensure we can capture *all* the Pareto optimal points and, consequently, our desired one.
>
> > Can you better clarify the potential challenges arising when optimizing Equation 24 with respect to classical differentiable recourse?
>
> When optimizing Equation (24) generally, there is no need to predefine a feasible set of features. However, if we want to restrict the optimization to a specific subset of variables that we intend to change, we can select them and solve the optimization accordingly. This approach contrasts sharply with causal algorithmic recourse, where one must *actively search* for the optimal subset of variables to intervene on from all feasible options. Their method relies on combinatorial optimization; however, when we fix a set of feasible variables in our method, it essentially reduces to a hyperparameter choice. In other words, if one claims to solve causal algorithmic recourse while simply fixing the feature set instead of optimizing to identify the optimal variables for intervention, the solution lacks the clear intuition behind an interventional counterfactual.
>
> [1] Karimi et al., Algorithmic recourse under imperfect causal knowledge: a probabilistic approach. Advances in Neural Information Processing Systems, 33:265–277, 2020.

---

### Official Review · Reviewer_cSC4 · 2025-03-13

**Overall Recommendation:** 4

**Summary:**

This paper proposes a new and efficient method for backtracking counterfactuals using causal reasoning to develop the explanations. The paper provides an analysis of the method’s limitations, discusses the relationship to the literature, and provides experiments that show promising results of their techniques.

**Claims And Evidence:**

Yes, the paper does a good job explaining the methodology, relating it to other literature in the field, proving several of the attributes of the methodology – thus providing a theoretical basis for the research – and it gives a strong experimental analysis.

**Essential References Not Discussed:**

N/A. This paper does a thorough literature review.

**Experimental Designs Or Analyses:**

The experimental design for this paper is sound and is based on previous research by Karimi et al. This allows the paper to have a direct baseline to compare their results to which is an important feature. The paper suggests promising results that improve upon three different papers from the last 8 years for algorithmic recourse. They also run a sensitivity analysis on their results suggesting robustness of their methodologies.

**Methods And Evaluation Criteria:**

Yes. Though this is more of a theoretical paper, the experimental section also directly compares the algorithm proposed to three distinct previous algorithms in the literature on a benchmark dataset for algorithmic recourse.

**Other Comments Or Suggestions:**

N/A

**Other Strengths And Weaknesses:**

This paper is strong and provides many of the attributes of a good paper. It contributes to the literature directly and analyzes its claims rigorously.

**Questions For Authors:**

How would this algorithm perform on large state-of-the-art models?

**Relation To Broader Scientific Literature:**

This paper contributes to the broader scientific literature by providing a new method for backtracking counterfactuals explanations using a causal reasoning approach.

**Theoretical Claims:**

The paper makes several theoretical claims, e.g., theorem 5.1 discusses that backtracking counterfactuals generalize interventional counterfactuals as one of the solutions derived from the backtracking counterfactual formulation can be made the same as the interventional counterfactual. The proofs provided appear to be correct.

---

> ### Author Rebuttal · Authors · 2025-04-01
>
> Thank you for your kind feedback and positive review of our work. We appreciate your recognition of the theoretical contributions, experimental design, and overall rigor of our paper.
>
> Regarding your question on how our algorithm would perform on large state-of-the-art models, we acknowledge that this is an important aspect to consider. As mentioned in lines 422–428 of section 10 of our paper, we recognize the significance of evaluating our method on more complex models. While our current experiments were conducted on a benchmark dataset for algorithmic recourse, we plan to extend our investigations to include large-scale, state-of-the-art models in future work. A comprehensive evaluation in this context remains an exciting direction for our future research, and we are eager to explore how our approach performs in real-world, complex model scenarios.
>
> Thank you again for your encouraging feedback.

---

### Official Review · Reviewer_6rSy · 2025-03-14

**Overall Recommendation:** 2

**Summary:**

This paper proposes a new method for counterfactual explanation of model behavior for models that fall under the additive noise constraints. The new framework is based on backtracking counterfactuals, that find settings for exogenous variables that produce endogenous variables with a desired counterfactual value. They demonstrate in an experiment that their method for creating counterfactual inputs outperforms previous methods for causal algorithmic recourse on a small dataset.

**Claims And Evidence:**

Backtracking counterfactuals are argued to be more intuitive, and this is supported by deep tracking counterfactuals which seems clearly true because they don't relate x and x^CF.

It's really elegant how the set up simplifies to deep backtracking and counterfactual explanation as you vary the parameter! The whole thing seems really theoretically elegant and nice.

**Essential References Not Discussed:**

N/A

**Experimental Designs Or Analyses:**

The experiment is a very simple causal algorithmic recourse task. It fits the proposed method perfectly and adheres to the needed assumptions.

**Methods And Evaluation Criteria:**

The comparison between methods seems a bit fraught to me. Given that this is a single dataset with only five variables, I expected there to be more clear success criteria. Why is it bad that a method recommends reducing loan duration alone? I don't see why recommending lower loan duration and a lower loan is any less actionable or worse by some other metric.

Additionally, the robustness evaluations seem a bit ad hoc, why isn't there a comparison between different methods that systematically compares their robustness?

Compared to the rest of the paper, the experimental results and analysis seems like an after thought.

**Other Comments Or Suggestions:**

N/A

**Other Strengths And Weaknesses:**

The weakness of the paper is the assumptions needed for any of the results to go through. Assumptions of linearity and convexity in models is increasingly limiting in the modern age of machine learning. Especially when the motivation is algorithmic recourse, because the vast majority of AI models used in our society will be text/image/voice models that are highly non-linear and non-convex.

Also, even within the limited model class, the comparison to other methods seems limited in ways pointed out above.

The strength of this paper is that its well-written, easy to ready, and theoretically elegant.

**Questions For Authors:**

See methods and evaluation criteria.

**Relation To Broader Scientific Literature:**

Counterfactual explanation connects deeply to fairness and explainable AI, as we need counterfactual inputs to understand how models make decisions and what they would have done under different circumstances. However, the current paper presents a method that is limited to additive noise models, which limits its connections greatly.

**Theoretical Claims:**

I think I follow the 5.1 and 6.1 proofs in the paper, but I wouldn't say I am familiar enough with the material to check its correctness closely.

---

> ### Author Rebuttal · Authors · 2025-04-01
>
> Thank you for your insightful remarks regarding our paper. We truly appreciate your thoughtful feedback and look forward to exploring your suggestions further to enhance our work.
>
> > The current paper presents a method that is limited to additive noise models.
>
> First, we note that our method applies to any case where $\\mathbf{F}(\\cdot)$ is invertible- a condition that is more general than the Additive Noise Models (ANM). While we require the ANM assumption to prove our theorems and to establish the connection between our method and causal algorithmic recourse, the practical application of our method only necessitates that $\\mathbf{F}(\\cdot)$ is invertible.
>
> > Why is it bad that a method recommends reducing loan duration alone?
>
> For this question, please first refer to **Answer for Q1** of our rebuttal to Reviewer 7Lfk.
>
> Additionally, we observe similar behavior in other instances. For example, for a high-risk individual with attributes (male, 27, \\$14,027, 60), our solution for $\\lambda = 1.2$ is (male, 27, \\$13,149, 37.4). In comparison, the solution provided by Counterfactual Explanations and Causal Algorithmic Recourse is (male, 27, \\$14,027, 36.6), while the Deep Backtracking Explanations method yields (male, 31.9, \\$11,626, 41.1).
>
> For another high-risk individual with attributes (female, 24, \\$7,408, 60), our solution with $\\lambda = 1.2$ is (female, 24, \\$6,273, 30.9). In this case, the solution from Counterfactual Explanations and Causal Algorithmic Recourse is (female, 24, \\$7,408, 29.8), and the Deep Backtracking Explanations method produces (male, 30.1, \\$4380, 35.5).
>
> We believe these results underscore the practical relevance of our approach in delivering actionable insights.
>
> For your question about robustness, please refer to **Answer for Q3** and **Answer for Q4** of our rebuttal to Reviewer 7Lfk.
>
> > Assumptions of linearity and convexity in models is increasingly limiting in the modern age of machine learning.
>
> It should be mentioned that, to the best of our knowledge, Theorem 5.1 in our paper is the first result to relate backtracking and interventional counterfactuals, and it holds independent importance. Additionally, Theorem 6.1 is, as far as we know, the first result that relates causal algorithmic recourse with backtracking explanation methods.
>
> Assumptions of linearity and convexity in our models are necessary because we require the vector optimization (Equation 20) to be convex. This convexity ensures that by varying $\\lambda$, we can capture *all* Pareto optimal solutions, thereby guaranteeing that we also obtain the desired Pareto optimal solution in Equation 18. However, as mentioned in the paper (lines 295–305), even without any additional assumptions—relying solely on the ANM assumption—we can *always* find a better solution than causal algorithmic recourse among the Pareto optimal points of the vector optimization (Equation 20). We need the assumptions of linearity and convexity to ensure that we can *capture* this Pareto optimal point for some $\\lambda$. That said, many Pareto optimal points of a non-convex vector optimization can still be reached, and our required Pareto optimal solution (the solution of Equation 18) might be one of them. For example, consider [1, Figure 4.9]. In the figure, all Pareto optimal points are shown as a bold line. Although the problem is non-convex, we can capture $f_0(x_1)$ and $f_0(x_2)$ with $\\lambda_1$ and $\\lambda_2$, respectively; however, there is no value of $\\lambda$ that can capture $f_0(x_3)$. We know that one of the points on the bold line is our desired point (Equation 18) under the ANM assumption. The linearity and convexity assumptions ensure that we can capture all the Pareto optimal points and, as a result, our desired one.
>
>
> [1] Boyd, Stephen P., and Lieven Vandenberghe. Convex Optimization Book. Cambridge university press, 2004.

---

### Official Review · Reviewer_7Lfk · 2025-03-14

**Overall Recommendation:** 3

**Summary:**

The authors propose a new framework for counterfactual explanations based on backtracking counterfactuals by introducing an optimization problem that seeks the nearest possible input modification needed to achieve the desired counterfactual outcome while preserving the causal relationships encoded in the input variables. They evaluate their framework on a simulated setup (structured causal model within a bank’s high-risk detection module) to show how their counterfactual explanations can be more intuitive for users and practical for real-world applications.

## update after rebuttal

Since my questions have been addressed, I have raised my score accordingly.

**Claims And Evidence:**

The claim of the proposed framework being more intuitive and practical needs more analysis and evidence to be convincing (explained more in the later sections – see "Experimental Designs or Analyses").

**Essential References Not Discussed:**

The related work provides a sufficient foundation for understanding the approaches evaluated in the paper.

**Experimental Designs Or Analyses:**

The experiment design is well-structured as a simulated setup; however, the accompanying analyses lack sufficient depth. Section 8 primarily describes the simulated setup and presents only a single case scenario, reporting counterfactual explanation results from various approaches, including their own. However, it does not provide enough qualitative or quantitative metrics to substantiate claims such as those in lines 409-410, where the authors assert that their approach “finds a balance … offering a more intuitive and actionable explanation for the user.” Similarly, in section 8.3, while the experimental setup is sound, lines 427-428 just state that “the results remain stable” without providing sufficient comparative context with other methods. To properly assess the robustness of the explanation, the experiments should explore multiple cases within this setup – ideally extending to other causal graphs to demonstrate generalizability. At a minimum, the study should thoroughly test variations of the stated causal graph across multiple individual instances, incorporating more comprehensive qualitative and/or quantitative evaluations.

**Methods And Evaluation Criteria:**

The proposed method and evaluation criteria makes sense for the problem at hand.

**Other Comments Or Suggestions:**

My primary concern is the depth of the experimental analysis. If more clarity and detail are provided, I would be open to reconsidering my score.

**Other Strengths And Weaknesses:**

Strengths:
* The paper provides a thorough background and problem setting.

Weakness:
* The experiment analyses lack sufficient depth (see “Experiment Designs or Analyses” and “Questions For Authors” sections), making it less convincing in demonstrating the improvement of this approach compared to existing methods.

**Questions For Authors:**

* Q1: Lines 403-404: I am confused as to why “reducing the repayment duration” is not an actionable item? Does this not mean that the individual can maybe repay faster to be considered a low-risk?

* Q2: Lines 407: Can we quantify how much of a “significant departure” it is? What is considered significant here? Is it how many features are changed or does it also look into the sum of relative change of each feature? In both cases, the counterfactual explanations (Wachter et al.) approach seems to have the least change.

* Q3: Line 427: What are the results for other methods? Did they change, and if so, to what degree?

* Q4: Lines  430-431: To better assess the robustness and reliability of the method, additional cases with more instances are necessary to understand how consistently the method remains stable. Ideally, to evaluate its generalizability, the method should be tested on various causal graphs. This would help determine the extent to which the approximate causal functions hold, such as how many features can be causally connected while still maintaining reliable performance.

**Relation To Broader Scientific Literature:**

The proposed counterfactual explanation framework presents a promising approach to generating closer counterfactual explanations while maintaining causal relationships with input variables. This addresses the gap of ensuring stability in such explanations and can be explored further in more complex settings to strengthen its practical use.

**Theoretical Claims:**

There are a couple of theoretical claims (in Section 5 and 6), but they are backed by sound proofs.

---

> ### Author Rebuttal · Authors · 2025-04-01
>
> Thank you for your thorough analysis and constructive feedback on our paper. We appreciate the opportunity to clarify the points raised and to provide additional insights into our research.
>
> **Answer for Q1:** In our quest for inputs that are actionable for the user, it is crucial to take the user profile into account. When the user’s initial features are given by x = (female, 24, \\$4308, 48), this suggests that a 48-month loan repayment is appropriate for the user. However, if we adjust this feature vector solely by reducing the repayment duration, the repayment becomes significantly more challenging for the user, thereby making the explanation less actionable.
>
> To put it quantitatively, repaying \\$4308 over 48 months corresponds to a monthly payment of \\$89.75. Any explanation that deviates considerably from this monthly rate is less actionable. Counterfactual Explanations (Wachter et al., 2017) and Causal Algorithmic Recourse (Karimi et al., 2021) yield a monthly repayment of about \\$131.3 (i.e., \\$4308 divided by 32.8 months). In contrast, our solution results in a monthly repayment of \\$123.8 when λ = 1 (i.e., \\$4087 divided by 33 months) and \\$112.2 when λ = 1.2 (i.e., \\$3736 divided by 33.3 months). Thus, while Counterfactual Explanations and Causal Algorithmic Recourse increase the monthly repayment by 46.3%, our approach leads to increases of 37.8% and 25.0% for λ = 1 and λ = 1.2, respectively, making our recommendation more actionable for the user.
>
> **Answer for Q2:** According to our optimization formulation (see Equation 26), by “significant departure” from the input we refer to the L1 norm difference, namely, $\left\| \mathbf{x} - \mathbf{x}^{\mathrm{CF}} \right\|_1$. This metric captures both the number of features modified and the aggregate relative change across those features. As you correctly mentioned, the Counterfactual Explanations method just minimizes this L1 norm, resulting in the smallest overall deviation from the original input. However, as highlighted in the Problem Definition point 3 (line 102), it is also essential to incorporate causal relationships among the input features. Consequently, our solution seeks an alternative that is not only close to the original observation in terms of the L1 norm but is also consistent with the underlying causal graph, thereby providing an actionable insight for the user.
>
> **Answer for Q3:** The following results demonstrate the performance of the Deep Backtracking Explanation method when identical noise is introduced:
>
> (female, 27.17, \\$2696, 35.8)
>
> (female, 27.14, \\$2752, 35.7)
>
> (female, 27.18, \\$2831, 35.6)
>
> As shown, the deviations are minimal in this method.
>
> Given that the causal algorithmic recourse method employs combinatorial optimization—which is typically sensitive to noise—we initially expected the introduction of noise to yield more dramatic changes in this method.
>
> It is important to note that when the counterfactual explanation only modifies the sink nodes of our causal DAG (like the example in our paper), the outcomes of the counterfactual explanation and the causal algorithmic recourse methods become identical. This is because interventions on these sink nodes are sufficient for interventional counterfactuals in causal algorithmic recourse optimization. In such cases, adding noise is less likely to alter the set of required interventions, so causal algorithmic recourse would be more robust to noises.
>
> However, as the added noise alters the causal graph, the solution of the causal algorithmic recourse shifts slightly. Under the same noise conditions, the new solutions of causal algorithmic recourse are:
>
> (female, 24, \\$4353, 32.7)
>
> (female, 24, \\$4192, 32.9)
>
> (female, 24, \\$4432, 32.6)
>
> Although these shifts are not dramatic, we anticipate that in more complex cases—where combinatorial optimization plays a larger role—the impact of noise could be more pronounced.
>
> **Answer for Q4:** For a high-risk individual with attributes (male, 23, \\$15,672, 48), our solution for explaining the model's behavior and providing actionable insights—using $\\lambda = 1.2$—is (male, 23, \\$15,116, 33.7). Under the same noise conditions described in our paper, our method yields the following results:
>
> (male, 23, \\$15,253, 33.6)
>
> (male, 23, \\$15,055, 33.8)
>
> (male, 23, \\$14,953, 33.9)
>
> For another high-risk individual with attributes (female, 24, \\$7,408, 60), our solution with $\\lambda = 1.2$ is (female, 24, \\$6,273, 30.9). When the same noise is applied, the method produces:
>
> (female, 24, \\$6,214, 31.0)
>
> (female, 24, \\$6,504, 30.7)
>
> (female, 24, \\$6,371, 30.8)
>
> Similarly, for another high-risk individual (male, 27, \\$14,027, 60), our solution for $\\lambda = 1.2$ is (male, 27, \\$13,149, 37.4). Under identical noise conditions, our method yields:
>
> (male, 27, \\$13,077, 37.5)
>
> (male, 27, \\$12,984, 37.6)
>
> (male, 27, \\$13,274, 37.3)
>
> These results underscore the stability of our approach across different input instances.

---

### Decision · Program_Chairs · 2025-05-01

**Decision:**

Accept (poster)

**Comment:**

This paper introduces a new framework for counterfactual explanations based on backtracking counterfactuals, aiming for more intuitive and practical explanations by preserving causal relationships. cSC4 and 7Lfk (after rebuttal) highlight the promising nature of the approach and its theoretical basis (cSC4, 6rSy), with 7Lfk noting the thorough background. However, a significant weakness identified by multiple reviewers (7Lfk, 6rSy, Fw7M) is the limited experimental evaluation, with concerns about the depth of analysis, the use of a single simulated dataset (7Lfk, 6rSy, Fw7M), and a lack of robust comparisons and statistical guarantees (6rSy, Fw7M). Fw7M also raises concerns about the justification of the proposed objective function and the strong assumptions underlying the theoretical claims. Overall, while the core idea is seen as interesting and theoretically elegant, the reviewers generally express reservations about the strength of the empirical evidence and the scope of the evaluation, requiring more substantial experimental validation to fully support its claims. The rebuttal improved Reviewer 7Lfk's score, but the fundamental concerns of other reviewers regarding the experimental rigor and justification of the method were not fully addressed.